# The Impact of Climate Change on the Water Systems of the Yesil River Basin in Northern Kazakhstan

**Anuarbek Kakabayev** [1], **Baurzhan Yessenzholov** [1], **Abilzhan Khussainov** [1], **Javier Rodrigo-Ilarri** [2,*], **María-Elena Rodrigo-Clavero** [2], **Gulmira Kyzdarbekova** [1] and **Gulzhan Dankina** [1]

1   Department of Mining, Construction and Ecology, Sh. Ualikhanov Kokshetau University, Abai Str. 76, Kokshetau 020000, Kazakhstan; anuarka@mail.ru (A.K.); e_baur_1985@mail.ru (B.Y.); abilzhan.khusainov@mail.ru (A.K.); gulmira.kyzdarbekova.80@mail.ru (G.K.); dankina_g_88@mail.ru (G.D.)
2   Instituto de Ingeniería del Agua y del Medio Ambiente (IIAMA), Universitat Politècnica de València, 46022 Valencia, Spain; marodcla@upv.es
*   Correspondence: jrodrigo@upv.es

**Abstract:** The geographical location of Kazakhstan, situated in the central part of the Eurasian continent, has played a crucial role in shaping a distinctly continental climate. This positioning has led to Kazakhstan facing a significant challenge in terms of water resource availability. The country's water resources are highly vulnerable to the dual pressures of climate change and human activities. It is noteworthy that the Yesil River basin is the sole region within Kazakhstan's borders where water resources are predominantly generated, while all other river basins experience substantial outflows beyond the nation's boundaries. This research undertaking involves a comprehensive analysis of long-term climatic data collected from meteorological stations located within the confines of the Yesil basin in Northern Kazakhstan. Additionally, the study encompasses the computation of water consumption and annual runoff within this region. Historical meteorological observations spanning from 1961 to 2020 reveal notable trends. Most significantly, a 1.2 °C increase in temperature is observed during the spring season. Winters have also become relatively milder and warmer, particularly towards the end of February, where temperatures have shifted from −16.2 °C in the first 30-year period to −14.6 °C in the second period. These findings underscore the ongoing climatic changes within the region, with significant implications for the management and sustainability of water resources in Kazakhstan.

**Keywords:** climate change; water consumption; annual runoff; water resources sustainability

## 1. Introduction

The escalating growth of the global population and the increased demand for conventional energy resources such as oil, natural gas, and coal have jointly contributed to the observable phenomenon of climate change. In 1993, the World Meteorological Organization (WMO) released its first-ever State of the Climate report, offering insights into expected climate change trends [1]. Subsequent annual reports on the global climate have consistently emphasized critical indicators within the climate system. These indicators encompass greenhouse gas concentrations, rising land and ocean temperatures, sea level elevation, ice sheet melting, glacier retreat, and an upsurge in extreme weather events. Of notable concern, the average annual global temperature in the year 2021 registered at 1.11 ± 0.13 °C above the average temperature recorded during the pre-industrial period spanning from 1850 to 1900 [2]. Although this anomaly is somewhat less pronounced than in recent years, it remains a matter of significance. It is paramount to emphasize that the seven years spanning from 2015 to 2021 collectively stand as the warmest years documented throughout the entire history of climate observations.

In 2022, the annual global average concentration of carbon dioxide in the atmosphere increased to 417.1 ± 0.1 ppm, representing a 50% rise from pre-industrial levels. The

global mean tropospheric methane levels were 165% higher than their pre-industrial levels, and nitrous oxide concentrations were 24% above pre-industrial levels [3]. Kazakhstan's geographical location is at the intersection of two continents, Europe and Asia, spanning from 45 to 87 degrees east longitude and 40 to 55 degrees north latitude. This location places Kazakhstan equidistant from the Pacific and Atlantic Oceans as well as from the Indian and Arctic Oceans. The country's distance from major bodies of water, combined with its vast territory, significantly influences its climate. Kazakhstan falls within the southern latitudes of the temperate zone, resulting in a dry and markedly continental climate. As one moves from west to east along the latitudinal axis, the continentality of the climate intensifies. Due to its geographical position, Kazakhstan experiences distinct seasonal variations, with four distinct seasons clearly manifested [4].

In 2021, Kazakhstan experienced a significant rise in the average annual air temperature, with an anomaly of +1.58 °C compared to the long-term average value for the period between 1961 and 1990, which was 5.4 °C. It is worth noting that this temperature anomaly was 0.34 °C lower than the anomaly observed in the previous year, 2020. This trend of increasing temperatures in the region has been ongoing since the 1960s, with each decade consistently warmer than the one before it. The most recent decade, spanning from 2012 to 2021, recorded an average annual air temperature of +6.61 °C. This figure surpassed the climatic norm by 1.19 °C, marking it as a record value for positive decadal anomalies [5]. To provide context, the warmest decade prior to this was from 2001 to 2010, with an anomaly of +1.09 °C [5]. Additionally, the most recent five-year period, covering 2017 to 2021, was characterized by exceptionally high temperatures, featuring an average annual air temperature of +6.69 °C. This temperature anomaly exceeded the climatic norm by 1.27 °C [5]. These findings emphasize the persistent warming trend in Kazakhstan, highlighting the importance of comprehending and addressing the consequences of such climate changes for the region and its environmental and societal systems [5].

In 2021, Kazakhstan witnessed a significant departure from the historical norm in terms of average annual precipitation, amounting to 85.5% of the long-term average, approximately 272 mm. It is crucial to note that the western, northern, central, and southern regions of the country, including areas such as Atyrau, Kostanay, North Kazakhstan, Akmola, Karaganda, Turkestan, Zhambyl, and Almaty, were notably impacted by a precipitation deficit, ranging from 20% to 40% below historical averages [5]. These deviations in precipitation patterns are indicative of the region's vulnerability to the effects of climate change. These changes in the climate system present significant challenges for Kazakhstan's natural ecosystems. One of the most pressing concerns stemming from these changes is the imminent scarcity of water resources, which is likely to have far-reaching consequences. Notably, the shortage of water resources is expected to have a particularly adverse impact on the agricultural sector, which heavily depends on consistent and ample water supplies. Additionally, these shifts in climate and water availability may have adverse effects on the elemental composition of the environment and the overall health of the population [6].

These observations emphasize the immediate need for proactive measures and strategies to address the multifaceted challenges posed by climate change in Kazakhstan. There is a particular emphasis on sustainable water resource management and bolstering the resilience of critical sectors such as agriculture and public health [5,6]. The future climate projections for Kazakhstan present a troubling outlook, indicating a persistent trend of increasing temperatures and heightened climate variability. These projections are based on average scenarios and consider the intricacies of climate modeling, as well as the uncertainties associated with changes in greenhouse gas (GHG) concentrations [7].

By 2030, it is anticipated that the average annual temperature in Kazakhstan will rise by 1.4 °C. This warming trend is expected to intensify over the coming decades, with estimates suggesting an increase of 2.7 °C by 2050 and a substantial 4.6 °C by 2085 [7]. This ongoing temperature rise is likely to result in a reduction in the number of frosty days, which can have various ecological and agricultural implications.

In terms of precipitation patterns, there are notable variations in the projections. By 2050, it is expected that Kazakhstan will witness an increase in winter precipitation of approximately 9% and a 5% increase in spring precipitation compared to historical norms [8]. Additionally, there is an anticipation of a heightened intensity and variability in precipitation events, which could have implications for water resource management and flood risks. However, it is essential to acknowledge that there exists significant uncertainty in precipitation projections. Some climate models suggest an increase in annual precipitation of 2% by 2030, 4% by 2050, and 5% by 2085. In contrast, other models predict a decrease in precipitation by 2085, with an average reduction of 11% [9]. This variability underscores the complexity of forecasting precipitation changes in a changing climate [3].

In an extreme scenario characterized by high greenhouse gas (GHG) emissions, there is a troubling projection that the wet zone will shift northward by 250–300 km by 2085 [10]. Under such circumstances, the northern regions of Kazakhstan may undergo a transition into semi-desert zones, a change that would carry profound ecological and societal implications. These climate projections underscore the imperative need for robust climate adaptation and mitigation strategies in Kazakhstan to address the challenges arising from increasing temperatures, shifting precipitation patterns, and the associated impacts on ecosystems, agriculture, and water resources [7]. Data from the United Nations Food and Agriculture Organization (FAO) highlights an intriguing paradox in Central Asian countries concerning water resources. On a per capita basis, the region appears to have a sufficient supply of water resources, averaging around 2.3 thousand $m^3$ per person. However, the real challenge in the region lies not in the absolute scarcity of water but rather in the inefficient and unsustainable use of these resources.

It has been recognized that regional climate models are essential for climate change projections and hydrologic modelling studies, especially in watersheds that are overly sensitive to changes in climate [11]. Addressing water-related challenges in Central Asia will require a multifaceted approach. This includes implementing sustainable water management practices, improving irrigation efficiency in agriculture, enhancing water infrastructure, and developing regional cooperation mechanisms to manage shared water resources effectively. Additionally, strategies for adapting to and mitigating the impacts of climate change will be crucial in ensuring water security in the region [12].

Taking into account everything said above, the general objective of this research is to study the impact of climate change in Northern Kazakhstan on the water systems of the Yesil basin. The following specific objectives are considered:

- Studying Temperature and Precipitation Dynamics: Monitoring the dynamics of temperature changes and precipitation patterns is fundamental to understanding the local climate trends. These data provide essential insights into how the climate in Northern Kazakhstan is evolving, which is critical for assessing its impact on water resources.
- Calculating Water Consumption: Calculating the water consumption of specific rivers in the Yesil basin, including the Yesil, Akkanburlyk, Imanburlyk, and Zhabai rivers, is essential for water resource management. This information helps in evaluating the availability and sustainability of water resources for various uses, including agriculture, industry, and domestic needs.
- Analyzing Population Dynamics: Examining changes in the population of the Yesil River basin is relevant as it can provide insights into the increasing demand for water resources due to demographic shifts. This analysis helps in projecting future water needs and planning for sustainable resource management.
- Considering Climate and Hydrological Scenarios: Assessing climate and hydrological indicators under different scenarios is crucial for understanding potential future changes. These scenarios allow for the development of adaptive strategies to address the impacts of climate change on water systems.

The scientific novelty of this research is rooted in its dedicated focus on an underrepresented region, Northern Kazakhstan, and its unwavering commitment to addressing the

critical challenges of climate change and water scarcity in this specific area. Although global climate change and its impacts on aquatic ecosystems have been extensively studied, the specific regional consequences can vary significantly. By directing our attention to Northern Kazakhstan, we provide valuable local insights that can inform regional policymakers, researchers, and stakeholders.

Recent and relevant research underscores the significance of our work within the context of Kazakhstan. The awareness of global climate change and its effects on aquatic ecosystems has garnered widespread attention [13–17]. In Kazakhstan, recent studies have shed light on the tangible consequences of climate change and water scarcity [18–21]. However, investigations of this nature have been notably lacking in the northern regions of Kazakhstan [22–24]. Our study effectively bridges this research gap by focusing on Northern Kazakhstan, where inquiries regarding climate change and water scarcity have been conspicuously limited. Our research encompasses the study of climate, hydrology, and population growth dynamics in the Yesil River basin, along with an assessment of water consumption. These findings enable us to adapt the river system to climate change and implement preventive measures.

## 2. Materials and Methods

The geographic area under examination falls within the classification known as the Kazakhstani type, as categorized by B.D. Zaikov [25], and is characterized as a region exclusively reliant on snow-based water supply, following M.I. Lvovich's classification [26]. In this region, the annual river flow is predominantly generated during the spring flood season, contributing a significant portion of the yearly water flow, typically ranging from 95% to 98%. The Yesil Water Management Basin (WMB) extends across three administrative regions in Kazakhstan: North Kazakhstan, the majority of the area north of the capital, Akmola, and a small segment of the Karaganda Region, specifically the upper reaches of the Yesil River. Surface water resources within the Yesil WMB primarily rely on the runoff of the Yesil River itself and the rivers situated in the Yertis–Yesil interfluve, including the Chaglinka, Kamysakty, Selety, and others. It is crucial to emphasize that the Yesil WMB is situated in an arid or inadequately moist zone, resulting in relatively limited surface water resources in the region [27].

The Yesil Water Management Basin (WMB) encompasses an area measuring 237,226 km$^2$. The primary watercourse within this region is the Yesil River, which receives contributions from several significant tributaries originating in the north, from the Kokshetau upland, and in the south, descending from the spurs of the Ulytau mountains. Notable tributaries of the Yesil River include Kalkutan, Zhabai, Terisakkan, Akkanburlyk, and Imanburlyk. Across the region under consideration, there are a total of 45 reservoirs. Lakes and reservoirs, comprising surface water bodies that vary significantly seasonally, play an essential role in the global water cycle due to their ability to hold, store, and clean water [28].

The long-term regulation of the Yesil River's flow is primarily facilitated by two reservoirs: the Astana (Vyacheslav) Reservoir and the Sergeyevsk Reservoir. It is worth noting that the Yesil River serves as a left-bank tributary of the Yertis River, spanning a length of 2450 km, with a catchment area encompassing 177,000 km$^2$, of which 141,000 km$^2$ are considered active [29].

This study relies on data sourced from the State Hydrometeorological Fund of the RSE "Kazhydromet", specifically monthly and annual records of air temperature and precipitation spanning the period from 1991 to 2020. The research methodology employs statistical analysis techniques applied to time series data. A key aspect of the study involves a comparative analysis of the statistical characteristics pertaining to average air temperature and precipitation across the water basin's territory for two consecutive periods: 1961–1990 and 1991–2020. The research was conducted through a structured series of phases, as outlined below. These structured phases ensured a systematic approach to the research, enabling a thorough analysis of the climate and water resource dynamics in the Yesil river basin in Northern Kazakhstan.

- Description of the study area: Initially, the study began with a detailed description of the geographical and environmental characteristics of the study area, which in this case is Northern Kazakhstan. This step served to establish the context for the research.
- Compilation of available data from existing weather stations: The next phase involved the collection and compilation of relevant meteorological data from pre-existing weather stations within the study area. This data likely included information such as temperature, precipitation, humidity, and other pertinent climatic variables.
- Calculations of water flow and runoff: Following the data compilation, calculations were carried out to determine water flow and runoff in the study area. This phase would involve the analysis of precipitation, evaporation, and other hydrological parameters to understand the dynamics of water flow within the region.
- Determination of monthly and annual averages: To gain a comprehensive understanding of the climate and hydrological patterns, the research included the determination of monthly and annual averages. This would allow for the identification of seasonal variations and long-term trends in the study area's environmental conditions.

Figure 1 illustrates a map detailing the geographical extent of the Yesil water basin.

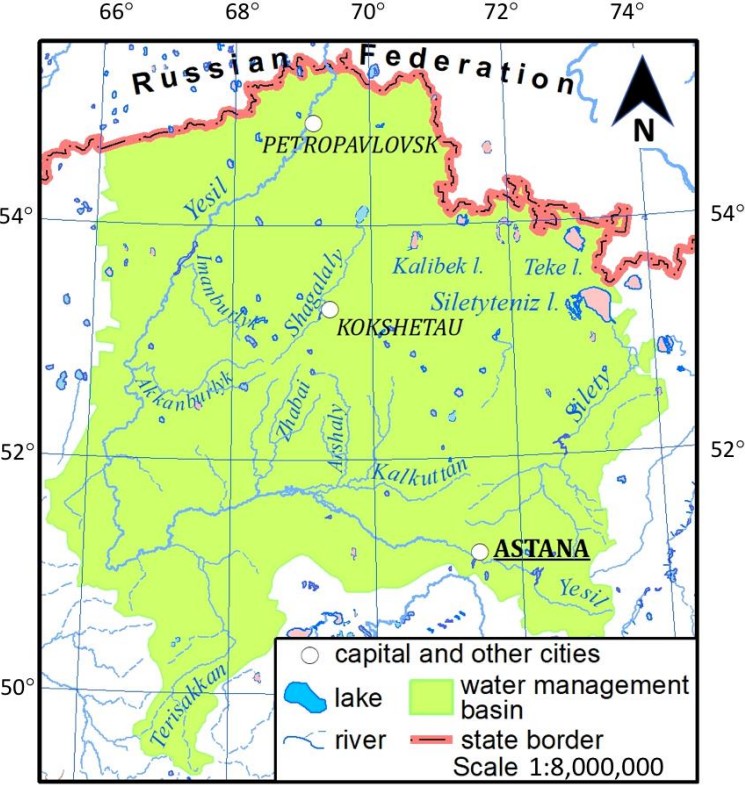

**Figure 1.** Map of the Yesil water basin.

In alignment with the criteria established by the World Meteorological Organization (WMO) in 2017, the calculation of average climatological data is performed over 30-year intervals. Hence, the climatological standard norms for this study are based on data collected from 1 January 1991 to 31 December 2020. Table 1 provides the exact coordinates of the 25 weather stations that played a crucial role in supplying the data used in this research.

**Table 1.** Location of the weather stations used in this study.

| Weather Station | Coordinates WGS84 | Absolute Height, m |
|---|---|---|
| Akkol | 52°0′0″ N, 70°56′0″ E | 383 |
| Arshaly | 50°50′3.49″ N, 72°10′18.11″ E | 426 |
| Astana | 50°50′3.49″ N, 72°10′18.11″ E | 349 |
| Atbasar | 51°49′0″ N, 68°21′0″ E | 302 |
| Balkashino | 52°31′3.1″ N, 68°45′9.41″ E | 398 |
| Blagoveshenko | 52°31′3.1″ N, 68°45′9.41″ E | 150 |
| Vozvishenko | 52°47′50.7″ N, 66°45′12.09″ E | 125 |
| Yereimentau | 51°37′0″ N, 73°6′0″ E | 397 |
| Yesil | 51°57′20″ N, 66°24′15″ E | 219 |
| Zhaksy | 51°54′0″ N, 67°19′0″ E | 386 |
| Zhaltyr | 51°37′25.47″ N, 69°49′58.93″ E | 304 |
| Zheleznodorozhny | 52°5′51″ N, 65°40′14″ E | 252 |
| Kyshkenekol | 53°38′3″ N, 72°20′17″ E | 137 |
| Kokshetau | 53°17′30″ N, 69°23′30″ E | 228 |
| Petropavl | 54°51′44″ N, 69°8′27″ E | 140 |
| Presnegorkovka | 54°29′59″ N, 65°45′56″ E | 160 |
| Ruzayeka | 52°49′22″ N, 66°57′5″ E | 226 |
| Saumalkol | 53°17′29.09″ N, 68°6′16.71″ E | 385 |
| Sergeyevka | 53°52′48″ N, 67°24′57″ E | 153 |
| IBMP * Burabay | 53°5′14″ N, 70°18′0″ E | 352 |
| Tainsha | 53°50′52″ N, 69°45′50″ E | 155 |
| Timiryazeva | 53°44′45.13″ N, 66°29′13.88″ E | 171 |
| Chkalovo | 53°37′31″ N, 70°25′50″ E | 165 |
| Shuchinsk | 52°56′0″ N, 70°12′0″ E | 393 |
| Yiavlenka | 54°20′37.5″ N, 68°27′30.78″ E | 113 |

* Integrated background monitoring post.

The measurement of water flow was calculated from the data obtained from the hydrological posts of the State Hydrometeorological Fund of the RSE "Kazhydromet". The water flow rate at the hydrological station was determined by multiplying the cross-sectional area in the hydrological station's alignment (S) by the average flow velocity (V):

$$Q = S \cdot V \left[ m^3/s \right]$$

RSE "Kazhydromet" provided average daily data on water consumption and the monthly and annual water consumption averages were calculated. The runoff volume (W), being the amount of water flowing from the catchment area per year, was determined using

$$W = Q \cdot T \left[ m^3 \right]$$

where Q is the average water consumption over the estimated time period, $m^3/s$, T is the number of seconds in the estimated time period. The calculated parameters (average long-term water consumption) and the values of the annual runoff of a given water security were obtained from the series given for a multi-year period.

## 3. Results

### 3.1. Weather Stations Data Analysis

Based on data gathered from 25 meteorological stations within the Yesil basin, an analysis of average monthly and annual air temperature and precipitation was conducted for the period from 1991 to 2020. The study unveiled significant variations in temperature and precipitation patterns across the region.

In January, the average monthly temperature ranged from −17.3 °C in Kishkenekol, located in the upland areas, to −14.3 °C in Kokshetau. In July, the average monthly temperature exhibited variability, ranging from 18.2 °C in Balkashino to 20.6 °C in Astana, situated within the Yesil region. Over the span of 30 years, the long-term annual temperature

averages ranged from 1.7 °C in Balkashino to 3.9 °C in Astana. This data indicates that within the territory of the Yesil basin, the amplitude of temperature fluctuations amounted to 2.2 °C, as summarized in Table 2.

**Table 2.** Average monthly and annual air temperature (°C) for 1991–2020.

| Weather Station | January | February | March | April | May | June | July | August | September | October | November | December | Yearly Average |
|---|---|---|---|---|---|---|---|---|---|---|---|---|---|
| Akkol | −15.8 | −14.6 | −7.2 | 5.0 | 13.1 | 18.2 | 19.2 | 17.5 | 11.0 | 3.8 | −6.4 | −13.2 | 2.6 |
| Arshaly | −15.4 | −14.6 | −7.0 | 5.4 | 13.4 | 18.6 | 19.6 | 18.1 | 11.6 | 4.0 | −6.2 | −13.0 | 2.9 |
| Astana | −14.5 | −13.6 | −6.0 | 6.5 | 14.5 | 19.6 | 20.6 | 19.1 | 12.6 | 5.0 | −5.2 | −12.0 | 3.9 |
| Atbasar | −16.9 | −15.8 | −8.3 | 4.9 | 13.7 | 18.8 | 19.9 | 18.3 | 11.7 | 4.1 | −6.6 | −14.0 | 2.5 |
| Balkashino | −16.4 | −15.1 | −7.9 | 3.6 | 12.3 | 17.2 | 18.2 | 16.4 | 10.2 | 3.1 | −7.1 | −13.9 | 1.7 |
| Blagoveshenko | −16.5 | −15.1 | −7.4 | 4.3 | 13.2 | 18.3 | 19.6 | 17.5 | 11.2 | 4.0 | −6.4 | −13.6 | 2.4 |
| Vozvishenko | −17.3 | −15.7 | −8.1 | 4.2 | 13.0 | 18.1 | 19.4 | 17.2 | 10.8 | 3.6 | −7.2 | −14.1 | 2.0 |
| Yereimentau | −14.7 | −13.7 | −6.4 | 5.4 | 13.1 | 18.4 | 19.6 | 18.1 | 11.7 | 4.4 | −6.1 | −12.2 | 3.1 |
| Yesil | −15.5 | −14.5 | −6.8 | 5.9 | 14.5 | 19.5 | 20.6 | 19.0 | 12.3 | 4.7 | −5.7 | −12.9 | 3.4 |
| Zhaksy | −15.8 | −14.9 | −7.8 | 4.7 | 13.6 | 18.7 | 19.7 | 18.3 | 11.8 | 4.1 | −6.5 | −13.2 | 2.7 |
| Zhaltyr | −16.0 | −15.0 | −7.5 | 5.5 | 14.1 | 19.3 | 20.2 | 18.6 | 12.1 | 4.5 | −6.0 | −13.2 | 3.1 |
| Zheleznodorozhny | −16.1 | −15.2 | −7.9 | 4.9 | 14.0 | 19.3 | 20.4 | 18.8 | 11.1 | 4.2 | −6.2 | −13.5 | 2.8 |
| Kyshkenekol | −17.3 | −15.5 | −7.6 | 5.0 | 13.5 | 19.0 | 20.3 | 18.1 | 11.5 | 4.0 | −6.8 | −14.0 | 2.5 |
| Kokshetau | −14.3 | −13.1 | −5.7 | 5.1 | 13.5 | 18.6 | 19.6 | 17.9 | 11.8 | 4.7 | −5.5 | −11.8 | 3.4 |
| Petropavl | −16.5 | −14.8 | −7.0 | 4.5 | 13.3 | 18.1 | 19.5 | 17.3 | 11.0 | 4.0 | 6.6 | −13.7 | 2.4 |
| Presnegorkovka | −16.3 | −15.1 | −7.7 | 4.2 | 13.3 | 18.3 | 19.8 | 17.7 | 11.3 | 3.9 | −6.2 | −13.6 | 2.5 |
| Ruzayeka | −15.6 | −14.7 | −7.3 | 4.8 | 13.7 | 18.7 | 19.8 | 18.0 | 11.6 | 4.1 | −6.2 | −13.2 | 2.8 |
| Saumalkol | −15.2 | −14.0 | −6.8 | 4.4 | 13.0 | 17.7 | 18.7 | 17.0 | 10.9 | 3.8 | −6.5 | −13.0 | 2.5 |
| Sergeyevka | −16.2 | −14.9 | −7.3 | 4.7 | 13.7 | 18.6 | 19.8 | 17.8 | 11.5 | 4.3 | −6.3 | −13.4 | 2.7 |
| IBMP * Burabay | −14.5 | −13.0 | −5.9 | 4.9 | 12.5 | 17.5 | 18.6 | 16.9 | 10.7 | 3.7 | −6.1 | −12.0 | 2.8 |
| Tainsha | −15.6 | −14.1 | −6.4 | 5.4 | 13.6 | 18.7 | 19.8 | 17.8 | 11.5 | 4.3 | −6.1 | −12.7 | 3.0 |
| Timiryazeva | −16.1 | −15.0 | −7.5 | 4.7 | 13.7 | 18.8 | 19.9 | 18.0 | 11.6 | 4.2 | −6.2 | −13.3 | 2.7 |
| Chkalovo | −15.0 | −13.7 | −6.0 | 5.8 | 13.7 | 18.8 | 19.8 | 17.8 | 11.6 | 4.5 | −5.9 | −12.2 | 3.3 |
| Shuchinsk | −15.9 | −14.8 | −7.4 | 4.2 | 12.3 | 17.4 | 18.5 | 16.6 | 10.4 | 3.3 | −7.0 | −13.5 | 2.0 |
| Yiavlenka | −15.8 | −14.9 | −6.6 | 4.9 | 13.5 | 18.4 | 19.4 | 17.4 | 11.3 | 4.0 | −6.3 | −13.2 | 2.7 |
| Average | −15.8 | −14.6 | −7.1 | 4.9 | 13.4 | 18.5 | 19.6 | 17.8 | 11.8 | 4.1 | −5.8 | −13.1 | 2.7 |
| min | −16.9 | −15.8 | −8.3 | 3.6 | 12.3 | 17.2 | 18.2 | 16.4 | 10.2 | 3.1 | −6.8 | −14.0 | 1.7 |
| max | −14.3 | −13.0 | −5.9 | 6.5 | 14.5 | 19.6 | 20.6 | 19.1 | 12.6 | 5.0 | −5.2 | −11.8 | 3.9 |
| Amplitude | 2.6 | 2.8 | 2.4 | 2.9 | 2.2 | 2.4 | 2.4 | 2.7 | 2.4 | 1.9 | 1.6 | 2.2 | 2.2 |

\* Integrated background monitoring post.

An analysis of meteorological data from selected weather stations, including Sergeevka, Ruzaevka, Astana, and Atbasar, over the last decade (2013–2022) provides valuable insights into temperature trends in the region. Notably, the data highlights significant variations in temperature patterns during this period.

Among the years under examination, 2018 stands out as the coldest year, while 2020 emerges as the warmest. Specifically, in 2018, the average annual air temperature registered a notable decrease of 1.3–1.6 °C compared to the long-term annual average. In sharp contrast, 2020 experienced a substantial increase in the average annual temperature, with a rise of 1.7–2.3 °C. It is worth noting that the remaining years within the study period consistently demonstrated warmer conditions compared to the long-term annual temperature.

This temperature trend is further elucidated in Figures 2–5, which offer a visual representation of the temperature fluctuations observed during the analyzed decade. These findings emphasize the dynamic nature of temperature changes in the region and underscore the significance of short-term variations within the context of long-term climate trends. The distribution of precipitation within the Yesil River basin displays significant variations and follows seasonal patterns. Average annual precipitation levels differ across the region, ranging from 284 mm in SMS Zheleznodorozhny to 458 mm in SMS Saumalkol. These disparities in annual precipitation underscore the localized differences in moisture availability.

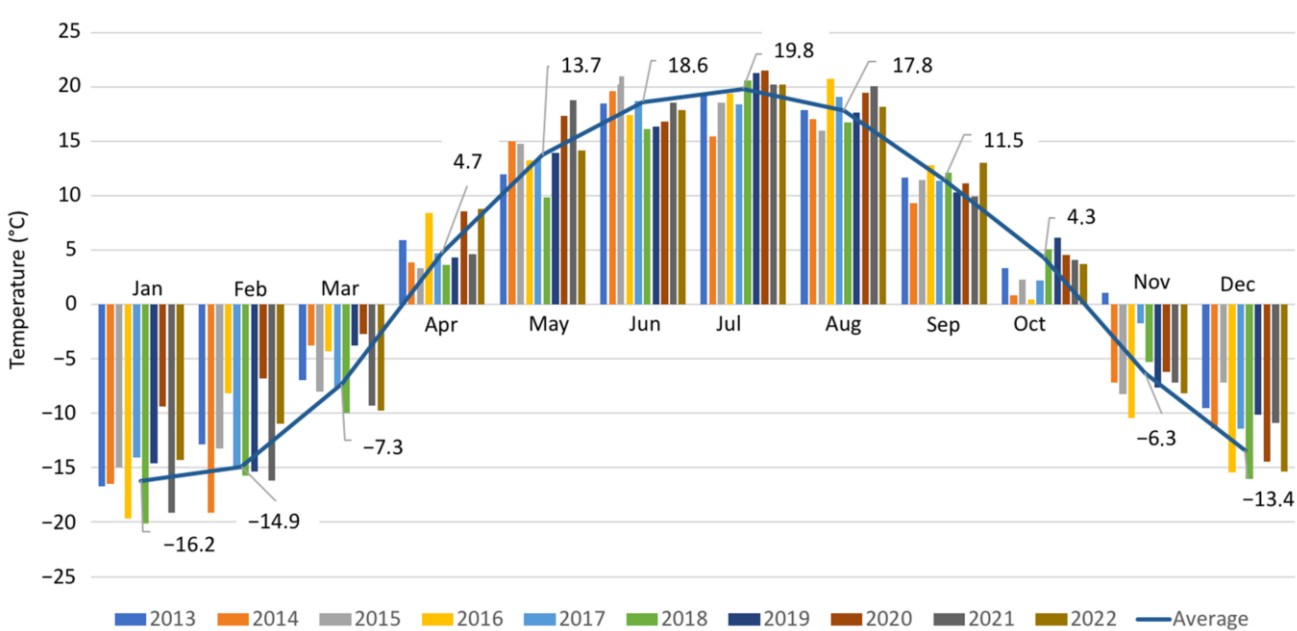

**Figure 2.** Average monthly and annual air temperature, according to the data of the Sergeyevka SMS for 2013–2022.

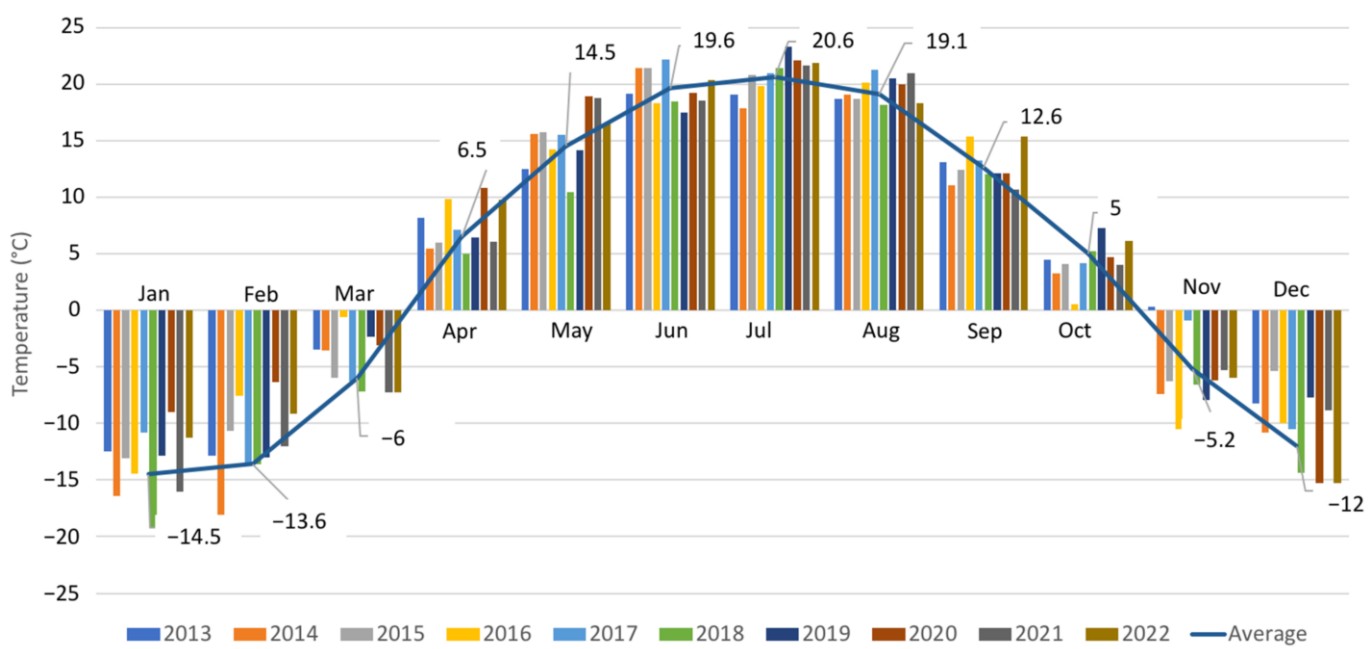

**Figure 3.** Average monthly and annual air temperature, according to the SMS of Astana for 2013–2022.

The majority of precipitation, averaging around 34%, takes place during the summer months. In some areas, like SMS Kokshetau, this summer precipitation can constitute as much as 49.8% of the total annual precipitation. On the other hand, winter experiences relatively lower levels of precipitation, ranging from 10.4% (SMS Burabay) to 24.8% (SMS Zhaksy) of the annual total. In contrast, spring and autumn exhibit a more even distribution of precipitation levels, with relatively equal amounts occurring during these transitional seasons. These distinct patterns of precipitation distribution throughout the year are crucial for comprehending the region's hydrology, ecosystem dynamics, and water resource management. They have implications for various sectors, including agriculture, water supply, and environmental sustainability, underscoring the importance of accurate and localized meteorological data in planning and decision-making processes (Table 3).

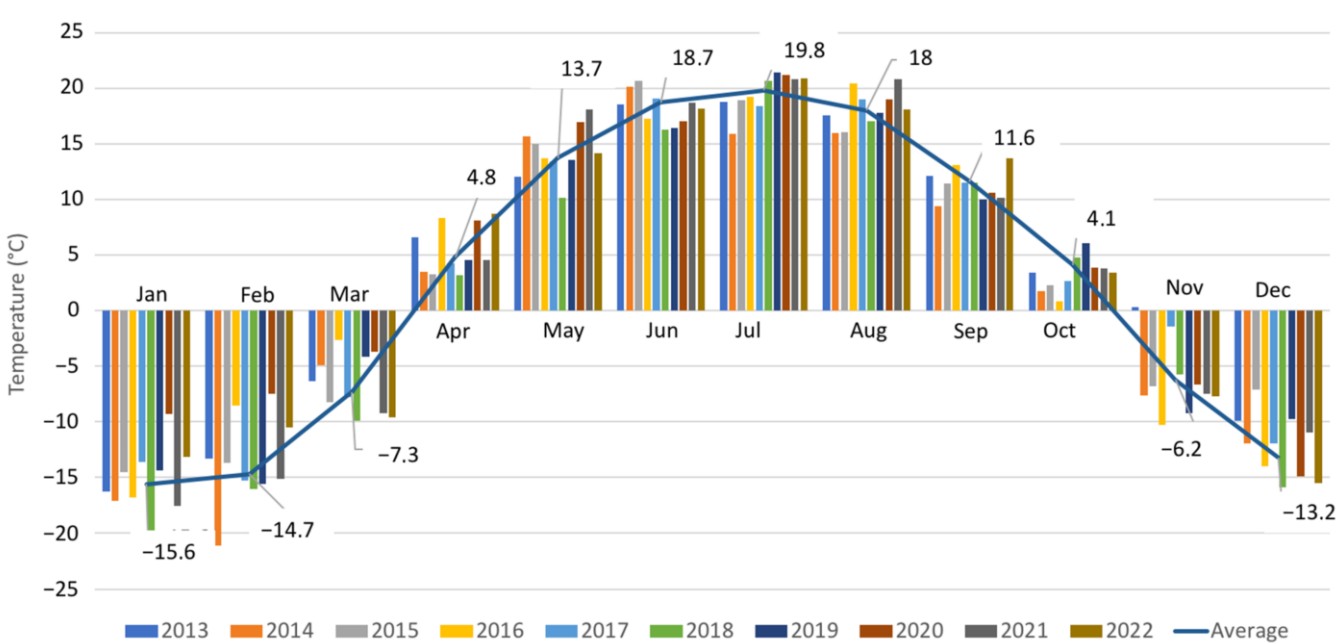

**Figure 4.** Average monthly and annual air temperature, according to the SMS Ruzayeka for 2013–2022.

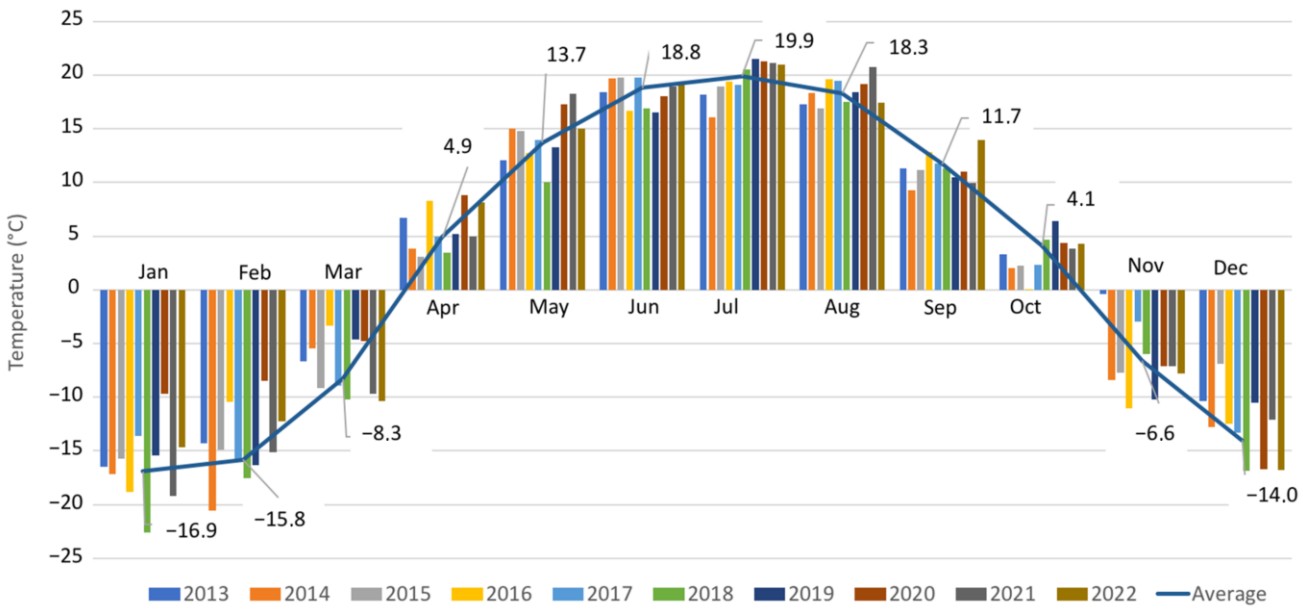

**Figure 5.** Average monthly and annual air temperature, according to the SMS Atbasar for 2013–2022.

Recent findings [30] highlight significant shifts in the 30-year climatic norms of air temperature between two distinct periods: 1961–1990 and 1991–2020. Notably, the latter 30-year span has exhibited a notable warming trend. Over this time frame, the average annual temperature has increased by 0.7 °C. Specifically, while it was at 1.9 °C in the period from 1961 to 1990, it rose to 2.7 °C in 1991–2020. This increase in average annual temperature is not limited to just the summer period; it encompasses all seasons. The most significant warming is observed during the spring season, with an increase of 1.2 °C. Additionally, winters have become relatively milder and warmer, particularly at the end of February, where temperatures decreased from −16.2 °C in the first 30-year period to −14.6 °C in the second.

**Table 3.** The average monthly and annual amount of precipitation (mm) for the period 1991–2020.

| Weather Station | January | February | March | April | May | June | July | August | September | October | November | December | Total Year |
|---|---|---|---|---|---|---|---|---|---|---|---|---|---|
| Akkol | 20 | 17 | 20 | 28 | 33 | 42 | 72 | 40 | 24 | 32 | 29 | 27 | 383 |
| Arshaly | 16 | 16 | 18 | 22 | 33 | 40 | 55 | 31 | 21 | 25 | 26 | 21 | 324 |
| Astana | 18 | 17 | 20 | 22 | 33 | 40 | 56 | 31 | 21 | 26 | 29 | 25 | 337 |
| Atbasar | 19 | 17 | 19 | 20 | 28 | 41 | 49 | 30 | 19 | 24 | 22 | 24 | 312 |
| Balkashino | 26 | 21 | 25 | 26 | 38 | 45 | 74 | 43 | 26 | 32 | 33 | 29 | 418 |
| Blagoveshenko | 16 | 13 | 18 | 24 | 30 | 46 | 61 | 45 | 33 | 29 | 24 | 18 | 355 |
| Vozvishenko | 12 | 11 | 13 | 20 | 28 | 49 | 65 | 47 | 26 | 26 | 22 | 15 | 334 |
| Yereimentau | 22 | 19 | 19 | 23 | 38 | 50 | 69 | 41 | 30 | 30 | 24 | 24 | 389 |
| Yesil | 13 | 12 | 15 | 19 | 33 | 36 | 52 | 29 | 17 | 22 | 22 | 18 | 287 |
| Zhaksy | 23 | 23 | 20 | 20 | 24 | 28 | 42 | 28 | 20 | 23 | 25 | 28 | 306 |
| Zhaltyr | 20 | 17 | 19 | 22 | 30 | 41 | 53 | 34 | 20 | 24 | 26 | 23 | 330 |
| Zheleznodorozhny | 16 | 17 | 21 | 18 | 31 | 31 | 40 | 27 | 17 | 24 | 22 | 21 | 284 |
| Kyshkenekol | 15 | 12 | 13 | 19 | 25 | 40 | 56 | 45 | 22 | 22 | 21 | 18 | 309 |
| Kokshetau | 12 | 11 | 13 | 19 | 26 | 43 | 72 | 44 | 23 | 22 | 18 | 15 | 319 |
| Petropavl | 19 | 16 | 20 | 24 | 33 | 45 | 69 | 45 | 31 | 30 | 30 | 25 | 387 |
| Presnegorkovka | 15 | 13 | 16 | 22 | 33 | 42 | 59 | 56 | 31 | 31 | 25 | 19 | 361 |
| Ruzayeka | 20 | 17 | 20 | 27 | 40 | 35 | 58 | 42 | 23 | 31 | 31 | 24 | 367 |
| Saumalkol | 25 | 21 | 25 | 30 | 39 | 47 | 81 | 52 | 34 | 40 | 36 | 31 | 458 |
| Sergeyevka | 19 | 17 | 22 | 26 | 37 | 41 | 62 | 52 | 29 | 29 | 26 | 23 | 383 |
| Burabay | 11 | 10 | 15 | 22 | 34 | 40 | 78 | 41 | 25 | 26 | 21 | 14 | 337 |
| Tainsha | 14 | 11 | 15 | 23 | 27 | 47 | 65 | 51 | 27 | 27 | 22 | 17 | 344 |
| Timiryazeva | 14 | 12 | 19 | 25 | 28 | 42 | 60 | 50 | 28 | 30 | 22 | 19 | 348 |
| Chkalovo | 13 | 10 | 15 | 22 | 29 | 46 | 67 | 50 | 23 | 25 | 22 | 17 | 339 |
| Shuchinsk | 15 | 12 | 16 | 22 | 32 | 42 | 82 | 42 | 26 | 26 | 21 | 17 | 352 |
| Yiavlenka | 17 | 13 | 19 | 29 | 34 | 48 | 75 | 47 | 30 | 29 | 22 | 19 | 382 |
| Avarage | 17.2 | 15.0 | 18.2 | 23.0 | 31.8 | 41.9 | 62.9 | 41.7 | 25.0 | 27.4 | 24.8 | 21.2 | 350 |
| min | 11 | 10 | 13 | 18 | 24 | 28 | 40 | 27 | 17 | 22 | 18 | 14 | 284 |
| max | 26 | 23 | 25 | 30 | 40 | 50 | 82 | 52 | 34 | 40 | 36 | 29 | 458 |
| Amplitude | 15 | 13 | 12 | 12 | 16 | 22 | 42 | 25 | 17 | 18 | 18 | 15 | 174 |

Although the average temperature in July experienced a slight decrease of 0.6 °C, dropping from 20.1 °C to 19.5 °C, there was an offsetting increase of 1 °C in August. These temperature fluctuations are critical indicators of the changing climate in the region, with potential implications for various aspects of the environment and human activities. Table 4 presents the statistical characteristics of the average air temperature in the territory of the Yesil water basin for the periods 1961–1990 and 1991–2020.

**Table 4.** Statistical characteristics of the average air temperature in the territory of the Yesil water Basin for the 1961–1990 and 1991–2020 periods.

| Period, Years | Climatic Norm for the Period | | | | ΔT (°C) |
|---|---|---|---|---|---|
| | 1961–1990 | | 1991–2020 | | |
| | Tsr (°C) | σ (°C) | Tsr (°C) | σ (°C) | |
| January | −16.6 | ±4.51 | −15.9 | ±3.98 | 0.7 |
| February | −16.2 | ±3.29 | −14.6 | ±3.86 | 1.5 |
| March | −8.7 | ±2.98 | −7.1 | ±3.45 | 1.6 |
| April | 4.0 | ±2.81 | 4.9 | ±2.71 | 0.9 |
| May | 12.4 | ±2.02 | 13.4 | ±1.75 | 1.0 |
| June | 18.2 | ±1.53 | 18.4 | ±1.88 | 0.2 |
| July | 20.1 | ±1.64 | 19.5 | ±1.67 | −0.6 |
| August | 16.8 | ±1.50 | 17.7 | ±1.75 | **1.0** |
| September | 11.2 | ±1.60 | 11.4 | ±1.50 | 0.2 |
| October | 2.2 | ±2.05 | 4.1 | ±1.92 | **1.9** |
| November | −6.7 | ±2.94 | −6.3 | ±3.85 | 0.4 |
| December | −13.3 | ±4.12 | −13.2 | ±3.34 | 0.1 |
| Winter | −15.4 | ±2.87 | −14.5 | ±2.76 | 0.8 |
| Spring | 2.6 | ±1.85 | 3.7 | ±1.89 | **1.2** |
| Summer | 18.4 | ±1.09 | 18.6 | ±1.06 | 0.2 |
| Autumn | 2.2 | ±1.51 | 3.0 | ±1.71 | 0.8 |
| Year | 1.9 | ±1.11 | 2.7 | ±0.92 | **0.7** |

Note: Tsr. is the average long–term deviation, σ is the standard deviation, Δ is the difference between the average long-term values. The statistically significant values of the difference according to Student's criterion t are highlighted in bold.

The analysis of temperature data within the Yesil River basin from 1976 to 2021 reveals a significant warming trend. Over this period, the average annual temperature has increased at a rate of 0.29 °C per decade, with the trend component contributing to 15% of the overall temperature variability. This warming trend has resulted in observable shifts in the timing of seasons. Notably, the onset of spring, a prolonged summer, and a later autumn have all been observed as occurring earlier. An illustrative example of this shift is the 1.9 °C increase in the average temperature of October from 1991 to 2020.

### 3.2. Statistical Analysis

Water management plays a crucial role in the planning and justification of economic activities within the basin. The basin generates a total of 2.2 km$^3$ of water, but various factors such as filtration, evaporation, sanitary and environmental releases, and unregulated runoff result in significant losses, leaving only 0.9 km$^3$ of available water resources for the needs of economic sectors [31].

In the case of the Yesil, Akkanburlyk, Imanburlyk, and Zhabai rivers, water consumption was calculated for the period from 2016 to 2020. Among these rivers, 2017 witnessed the highest water consumption levels, with significant increases compared to the long-term average. Specifically, water consumption on the Yesil River increased by 2.8 times, Akkanburlyk by 3.9 times, Imanburlyk by 5.3 times, and Zhabai by 11.7 times during this period. Conversely, the lowest water consumption was recorded in 2018, except for the Imanburlyk River. Notably, the Imanburlyk River experienced its minimum water consumption of 1.8 m$^3$/s in 2019, representing an 18.1% decrease compared to the long-term average, as shown in Figure 6.

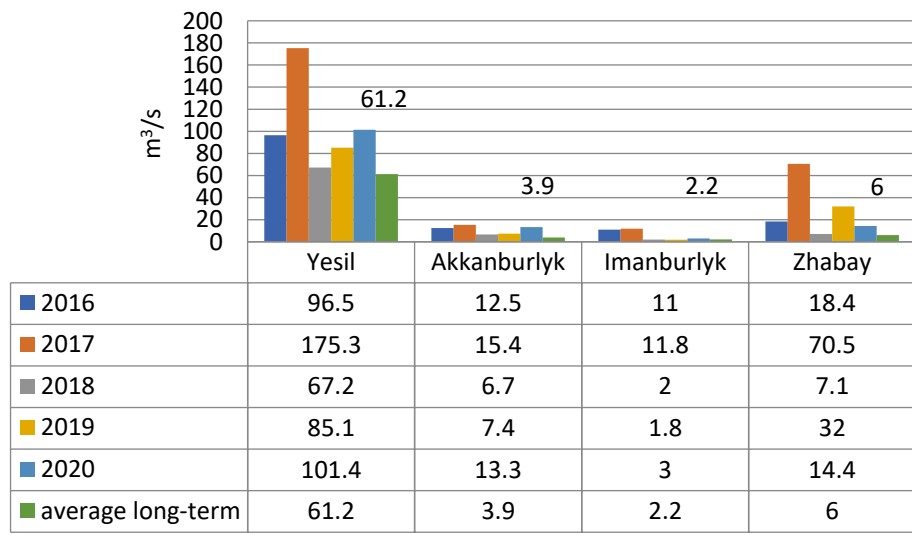

| | Yesil | Akkanburlyk | Imanburlyk | Zhabay |
|---|---|---|---|---|
| ■ 2016 | 96.5 | 12.5 | 11 | 18.4 |
| ■ 2017 | 175.3 | 15.4 | 11.8 | 70.5 |
| ■ 2018 | 67.2 | 6.7 | 2 | 7.1 |
| ■ 2019 | 85.1 | 7.4 | 1.8 | 32 |
| ■ 2020 | 101.4 | 13.3 | 3 | 14.4 |
| ■ average long-term | 61.2 | 3.9 | 2.2 | 6 |

**Figure 6.** Water consumption.

The data from the Sergeevka weather station provides valuable insights into the relationship between water consumption, average annual atmospheric air temperature, and annual precipitation. Notably, there is a moderate positive correlation (r = 0.31) between water consumption and average annual temperature, indicating that as temperatures rise, water consumption tends to increase. Conversely, there is an average negative correlation (r = −0.59) between the amount of annual precipitation and water consumption, suggesting that higher precipitation levels are associated with lower water consumption.

The long-term averages for air temperature and precipitation stand at 2.7 °C and 383 mm, respectively. However, in 2017, precipitation was notably lower, registering at 14.9% below the long-term average, while water consumption was significantly higher, reaching 286.4% above the long-term average. In contrast, 2018 saw air temperatures lower

than the long-term average by 1.3 °C, with precipitation levels aligning with the long-term average, and water consumption remaining at average levels.

These observations highlight that water consumption for the Yesil River at the Sergeevka gauging station during the analyzed period exceeded the average long-term levels. However, it is important to note that water consumption is not solely determined by climatic conditions, such as temperature and precipitation. Other factors, notably hydrogeological conditions, particularly groundwater, play a pivotal role in influencing the formation of water flow. These complex interactions underscore the multifaceted nature of water resource dynamics within the region. Table 5 presents the values of climatic and hydrological indicators of the Yesil River at the Sergeevka station.

**Table 5.** Climatic and hydrological indicators of the Yesil River at the Sergeevka station.

| Year | Water Consumption (m$^3$/s) | Average Annual Temperature (°C) | Annual Precipitation (mm) |
|---|---|---|---|
| 2016 | 96.5 | 2.8 | 435.4 |
| 2017 | 175.3 | 3.2 | 325.7 |
| 2018 | 67.2 | 1.4 | 376.2 |
| 2019 | 85.1 | 3.2 | 373.6 |
| 2020 | 101.4 | 5 | 416.5 |
| Correlation | | 0.31 | −0.59 |

The calculation of the annual flow of the Yesil, Akkanburlyk, Imanburlyk, and Zhabay rivers for the period from 2016 to 2020 is presented in Table 6. The results reveal a trend characterized by high water flow. During this period, the annual flow of these rivers consistently exceeded the long-term average, indicating a period of high water flow.

**Table 6.** Annual flow in the rivers of the Yesil basin (km$^3$/year).

| River | 2016 | 2017 | 2018 | 2019 | 2020 | Long-Term Average |
|---|---|---|---|---|---|---|
| Yesil | 3.0 | 5.5 | 2.1 | 2.7 | 3.2 | 1.9 |
| Akkanburlyk | 0.39 | 0.49 | 0.21 | 0.23 | 0.42 | 0.12 |
| Imanburlyk | 0.35 | 0.37 | 0.06 | 0.06 | 0.10 | 0.07 |
| Zhabay | 0.58 | 2.22 | 0.22 | 1.01 | 0.45 | 0.19 |

The peak annual runoff occurred in 2017, with Yesil recording 5.5 km$^3$, Akkanburlyk with 0.49 km$^3$, Imanburlyk with 0.37 km$^3$, and Zhabay with 2.22 km$^3$. In contrast, the lowest annual runoff figures were recorded in 2018.

This pattern suggests that in recent years, the average annual temperature within the Yesil River basin has experienced significant growth. This temperature increase is primarily attributable to the warming of winter, spring, and the extended duration of autumn. Despite these temperature changes, the annual runoff has remained consistently higher than the long-term average data. This hydrological trend underscores the complex relationship between climate variables and river flow dynamics, with the region experiencing both warming temperatures and robust river flows.

According to the Statistics Committee of the Ministry of National Economy of the Republic of Kazakhstan, a consistent upward trajectory in water consumption has been evident over the past decades. An examination of water consumption volumes from 2000 to 2021, as depicted in Figures 7 and 8, yields the following findings. In the Akmola region, water intake from rivers and reservoirs escalated from 42.9 million cubic meters to 55.7 million cubic meters, representing a 29.8% increase. Similarly, in the North Kazakhstan region, water intake surged from 22.6 million cubic meters to 33.9 million cubic meters, constituting a 50% rise. Within these figures, water consumption for industrial purposes exhibited growth, advancing from 8.7 million cubic meters to 11.7 million cubic meters,

marking a 34.4% increase in the Akmola region. In the North Kazakhstan region, the corresponding increase was even more pronounced, with water consumption for production needs climbing from 4.6 million cubic meters to 7.5 million cubic meters, signifying a 63% augmentation.

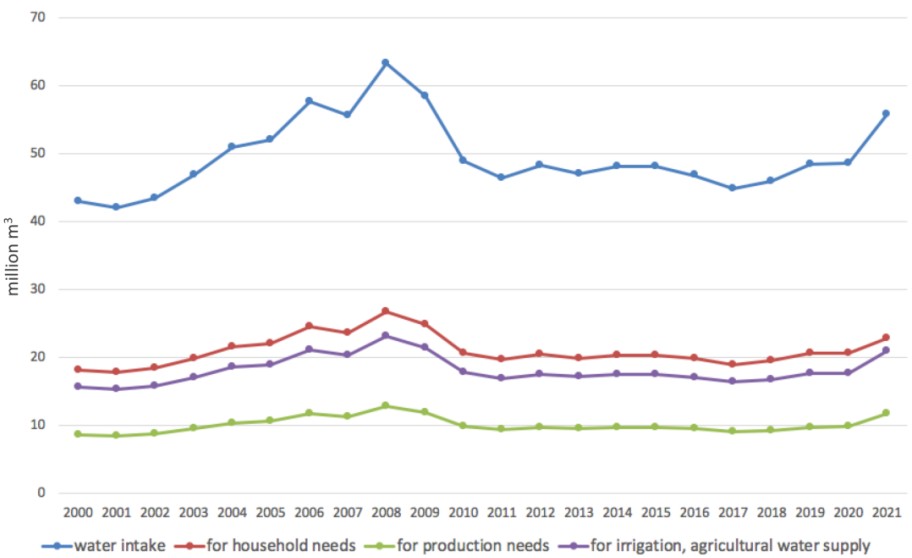

**Figure 7.** Water consumption by economic sectors for 2000–2021 in Akmola region.

For residential and potable water demands, the Akmola region witnessed a rise in water consumption, progressing from 18.2 million cubic meters to 22.8 million cubic meters, reflecting a 25.3% increment. In the North Kazakhstan region, a similar trend was observed, with water consumption for household and drinking purposes increasing from 9.6 million cubic meters to 13.6 million cubic meters, constituting a 41.6% surge. Furthermore, in the context of irrigation and agricultural water supply, water consumption showed an upward trajectory. In the Akmola region, water usage for these purposes increased from 15.6 million cubic meters to 20.9 million cubic meters, demonstrating a 33.9% growth. In the North Kazakhstan region, a comparable trend was evident, with water consumption for irrigation and agricultural water supply expanding from 8.2 million cubic meters to 12.2 million cubic meters, representing a 48.7% escalation.

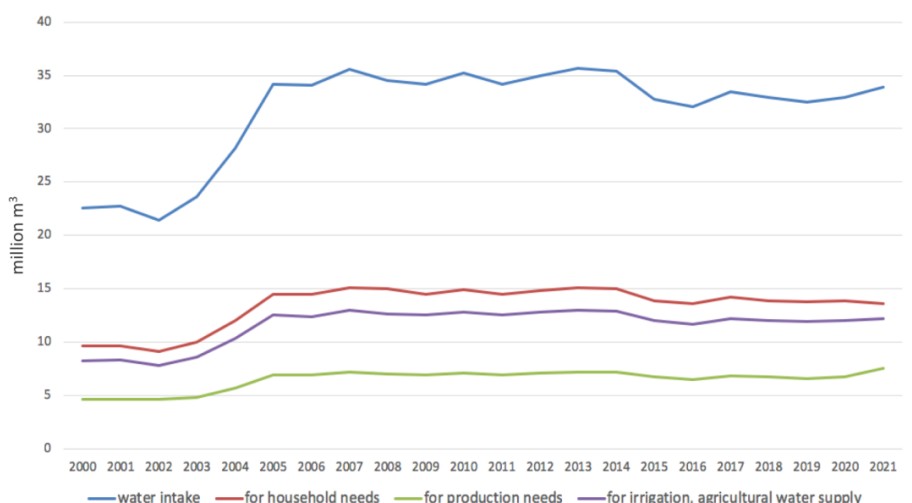

**Figure 8.** Water consumption by economic sectors for 2000–2021 in the North Kazakhstan region.

*3.3. Socioeconomic Factors*

Following data provided by the Kazakh National Committee on Water Resources, as of 2004, the primary water consumers within the Yesil River basin were distributed among various sectors as follows:

- Industry: 20.3%
- Irrigated agriculture: 22.0%
- Agricultural water supply: 14.5%
- Fisheries and other industries: 0.8%

These figures illustrate the allocation of water resources among different sectors in the region, emphasizing the significant roles played by industry and irrigated agriculture as the leading consumers of water. This distribution of water usage reflects the diverse demands on water resources within the basin's socio-economic activities, underlining the importance of efficient water management and conservation practices to ensure sustainable resource utilization.

In recent times, there has been a noticeable increase in the volume of water extraction to meet the demands of public utilities, primarily attributed to the rising water consumption in Astana. The municipal water supply in the region relies on both surface and underground water sources, with groundwater contributing a relatively small share, amounting to 8.5% of the total water intake. Within the territory under consideration, approximately 60% of the total water consumption for these needs is attributed to the residential sector. The remaining 40% is distributed among budget organizations and the local industrial sector.

As of 1990, the population in 1405 rural settlements numbered 945.0 thousand people. Most of this population, approximately 72.4%, was concentrated in central estates and district centers. By the end of 2004, the population in the territory had decreased to 802.4 thousand people, scattered across 1397 rural settlements, as depicted in Figure 9. These demographic shifts are essential considerations in assessing water demand and resource management within the region.

The Yesil River basin plays a pivotal role in providing the necessary water supply to residents of the Akmola and North Kazakhstan regions, as well as the city of Astana. Ensuring a full-fledged and high-quality water supply for both the population and the regional economy is a paramount concern for the government. To address this concern, an analysis of population growth within this area over the past two decades has been conducted.

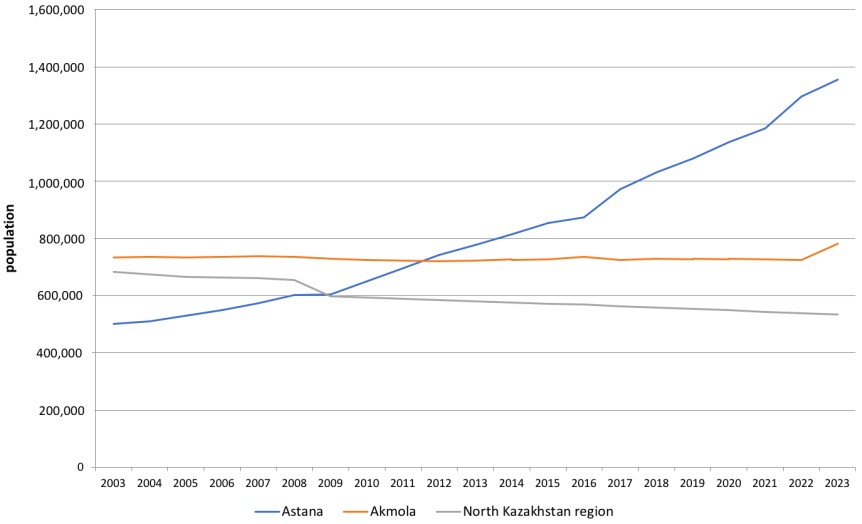

**Figure 9.** The dynamics of the population in the Yesil WMB.

In 2003, the population stood at 1,919,201 people, and by 2023, it had increased to 2,669,331 people, representing a significant growth of 139.1% over 20 years. A more detailed analysis by individual regions reveals distinct trends. In the Akmola region,

the population remained relatively stable until 2022, with a notable increase in 2023, reaching 780,716 people. In contrast, the North Kazakhstan region experienced a decline in population, decreasing from 682,148 people in 2003 to 534,108 people in 2023, a decrease of 148,040 people. This population decline is primarily attributed to the mass resettlement of ethnic Russians to their homeland in the Russian Federation, as the North Kazakhstan region is a border area.

Astana, on the other hand, witnessed substantial population growth, with a remarkable increase of 269.8% over 20 years, surging from 501,998 people to 1,354,507 people by 2023. This rapid population growth in the Yesil River basin has led to an increased demand for water resources, particularly in the domestic communal sector. Consequently, this places greater pressure on the main reservoirs, potentially leading to water scarcity issues in the future. Considering the interplay of factors such as climate change and population increase, it becomes imperative to promote the efficient and sustainable use of water resources and implement competent water resource management practices to address the evolving demands and challenges in the region.

## 4. Discussion

This research presents findings derived from historical temperature and precipitation data in the Yesil River basin. Additionally, it incorporates existing scientific literature on climate change forecasting in Northern Kazakhstan. Furthermore, the study includes an analysis of population dynamics, providing a comprehensive evaluation of the situation in the examined region. It is worth noting that the absence of certain data, such as greenhouse gas emissions from facilities in the Yesil River basin, restricted the ability to make precise climate change forecasts specific to this area. Consequently, broader studies encompassing Kazakhstan and Central Asia as a whole were leveraged for climate change predictions.

Global and regional climate change poses a significant threat to water supply in Central Asian countries, with Kazakhstan being particularly vulnerable. Projections indicate an increase in surface air temperatures, ranging from 0.8 to 1.5 °C throughout all months of the year. Furthermore, anticipated changes in average annual precipitation are expected, with alterations ranging from 1% to 3% by the year 2035 [32]. Based on climatic scenarios generated by five ISIMIP models, there is a consistent trend of rising average annual air temperatures in all catchment areas. These temperature increases are projected to range from 1.2 °C in the near future to as high as 6.4 °C in the distant future under various emission scenarios. Notably, the Zhabai River catchment area is expected to undergo the most substantial temperature rise, with projections of 3.9 °C under one scenario and 6.4 °C under another by the end of the century [33]. These climate changes have far-reaching implications for water resources in the region, necessitating the development of adaptive strategies to address the associated challenges.

The rise in temperature brings about various consequences, including a reduction in snow accumulation during the cold season and increased evapotranspiration in the summer. In specific regions, these changes may have a positive impact on agricultural production by extending the growing seasons [34]. Projections of the average monthly long-term flow of the Zhabai River indicate a substantial increase in the catchment area in the middle and far future for both RTCs, with the exception of May, where a 12% decrease is anticipated for RTK 4.5. In the near future, the river's flow is expected to decrease in most months according to RTK 4.5.

The absence of a consistent snow accumulation period throughout the year and a reduction in snow-derived water volume result in decreased water availability during the growing season and for irrigation purposes. Notably, the increase in temperature in snow-fed and glacial river areas significantly influences potential changes in river flow. This leads to significant seasonal fluctuations in river flow across many studied catchments, with a projected shift in peak flow occurring one month earlier, consistent with findings from other studies [35–37].

Climate change is expected to bring about an increase in average annual air temperatures, with a range of 1 °C to 7 °C, depending on emission scenarios. Projections also suggest an anticipated 10% increase in winter and other seasonal precipitation. Furthermore, there is an estimated daily rise of 0.2 mm in evaporation across all scenarios, leading to an elevated moisture deficit in the soil [38]. The significant temperature increase, especially during winter, spring, and autumn, results in heightened evaporation and altered freeze–thaw cycles. This leads to a decrease in winter precipitation, as snowfall declines, and an increase in rainfall, coupled with an overall reduction in total annual precipitation. While there is no unanimous consensus among researchers regarding the trend in total precipitation, observations from various authors do not strongly indicate a significant decrease in annual precipitation [39].

Recent assessments of average annual water consumption in selected rivers within the Yesil Water Management Basin (WMB) reveal a projected decrease in water runoff of 8.5%, 18.5%, and 19% for the periods spanning 2016–2045, 2036–2065, and 2071–2100, respectively [40]. These fluctuations can be attributed to the relative constancy of precipitation, despite a substantial increase in temperature and elevated evaporation in the region.

Hydrological forecasts indicate that, under both climatic scenarios RCP 2.6 and RCP 8.5, an increase in seasonal runoff is anticipated by 2040–2069 for the Irtysh, Tobol, Ural, and Yesil rivers. However, it is essential to note that forecasts concerning seasonal discharges exhibit a relatively lower level of reliability when compared to annual discharge projections [41].

## 5. Conclusions

In the context of global warming, it is imperative to conduct an in-depth investigation into the impact of climate change on the water systems of Northern Kazakhstan. This significance is underscored by the fact that the Yesil River serves as the primary water source for a population of 2.7 million people. The research employs "theoretical" methodologies, which encompass meta-analysis of published findings, analysis of extensive datasets obtained from long-term observations, and the utilization of simulation models to predict anticipated climate changes. The climate change dynamics within the Yesil water basin are characterized by a combination of increasing average monthly and seasonal surface air temperatures, while precipitation patterns remain relatively stable. Over the period from 1991 to 2020, monthly and seasonal air temperatures consistently exceeded those recorded in the preceding period from 1961 to 1990. The sole exception to this trend was July, which experienced a modest 0.6 °C decrease in temperature compared to the previous period. The most notable temperature increases were observed in October, rising by 1.9 °C, and in February and March, displaying increases of 1.5 to 1.6 °C.

Despite these temperature fluctuations, water consumption at the Sergeevka gauging station on the Yesil River consistently exceeded the long-term average from 2016 to 2020. This implies that water consumption does not always exhibit a direct correlation with temperature and precipitation levels. The formation of water flow is influenced by various factors, including hydrogeological conditions, notably the presence of groundwater. Certain hydrological forecasts, specifically those based on the ssp126 scenario, project a potential 18.5% reduction in the annual flow of the Yesil River for the period spanning from 2036 to 2065. The population within the Yesil River basin has experienced substantial growth, increasing by 39.1% from 2003 to 2023. This demographic expansion is most pronounced in the capital city of Astana, where the population surged by 169.8%. Because of this population growth, there is a continual rise in water consumption for household needs in the region. This trend underscores the critical importance of efficient and sustainable management of water resources in response to the evolving climate and expanding population.

**Author Contributions:** Conceptualization, B.Y. and A.K. (Abilzhan Khussainov); data curation, J.R.-I.; formal analysis, J.R.-I. and G.D.; funding acquisition, B.Y.; methodology, A.K. (Anuarbek Kakabayev); project administration, B.Y.; resources, A.K. (Abilzhan Khussainov); software, B.Y.; supervision, J.R.-I.

and M.-E.R.-C.; validation, M.-E.R.-C. and G.K.; visualization, J.R.-I. and M.-E.R.-C.; writing—original draft, B.Y.; writing—review and editing, J.R.-I. and M.-E.R.-C. All authors have read and agreed to the published version of the manuscript.

**Funding:** This research work was funded by the Science Committee of the Ministry of Science and Higher Education of the Republic of Kazakhstan (Grant No. AP13268760).

**Data Availability Statement:** Data are contained within the article.

**Conflicts of Interest:** The authors declare no conflict of interest.

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
