# Peer review of "The Impact of Climate Change on the Water Systems of the Yesil River Basin in Northern Kazakhstan"

_sustainability, doi:10.3390/su152215745_

Round 1

Reviewer 1 Report

Comments and Suggestions for Authors

This study extensively analyzes the long-term climate data obtained from meteorological stations in the Yesil Basin in northern Kazakhstan. Additionally, it includes calculations of water usage and annual runoff in the region. This research is significant and can be improved in the following aspects. Firstly, simplify the first three sentences of the abstract into one sentence to clarify the scientific problem and importance of the current study. Then, cite the latest similar research literature in the introduction section. In the methods section, it is suggested to outline and list the technical roadmap in bullet points. The conclusion section should be described in separate chapters. In conclusion, it is recommended to publish the study after the modifications are made.

Author Response

Dear reviewer #1,

I am pleased to submit the revised version of an original research article entitled “The impact of climate change on the water systems of the Yesil River basin in Northern Kazakhstan" for consideration for publication in Sustainability..

We have carefully updated the content of the manuscript following the comments of the four reviewers. This updated version of the manuscript takes into account all the suggestions made by the reviewers of the original manuscript. The new version of the modifies the requested sections of the original one. The manuscript has been reorganized and more references have been included. Some figures have been replotted as requested by some of the reviewers.

While the original version of the manuscript was 7975 words and included 28 references, this updated version is 9003 words long and includes 37 references. It is not possible to mark every change done in the English use inside manuscript, as it has been fully revised. Please activate the “track-changes” option in word to visualize all these changes.

We have provided answers to every reviewer’s comment which have been sent to each reviewer through MDPI online platform. Once all the comments of the reviewers have been addressed, we believe that this updated version of the manuscript fulfills all their requirements and is now appropriate for publication in Water.Please find below the answers to your specific comments. 

Comments reviewer #1

1.- Simplify the first three sentences of the abstract into one sentence to clarify the scientific problem and importance of the current study.

The full abstract section has been updated, as also requested by other reviewers.

2.- Cite the latest similar research literature in the introduction section.

New references have been provided. Besides, the introduction section has been revised.

3.- In the methods section, it is suggested to outline and list the technical roadmap in bullet points.

The outline has been included as requested

4.- The conclusion section should be described in separate chapters.

Discussion and conclusions are now separate chapters as requested.

Reviewer 2 Report

Comments and Suggestions for Authors

The paper entitled "The impact of climate change on the water systems of the Yesil River basin in Northern Kazakhstan" falls within the journal scope. The study is a very important exploration and has a strong guiding significance. Comments are given as follows:

1-     The introduction is too long. The section should be restructured in a way that clearly conveys the overall objective of the research. The goals should be restructured into general and specific objectives.

2-     Most paragraphs in the entire manuscript, especially in the introduction section, are either too lengthy or too short. Kindly consider at least 4 and at most 9 sentences in each paragraph. It is suggested to revise all through the manuscript.

3-      The novelty and practical applicability of this study should be highlighted more in the introduction section. The introduction section could be improved. Author should refer to these articles and several others relating to the subject matter from reputable journals to enrich and enhance the introduction and other sections of the manuscript. These are: https://doi.org/10.1007/s00704-023-04466-5

4-     The paper needs more effort in displaying the methodology as the applied is not so innovative. It lacks information of different indices used in this research. This section should be re-organized, more specific and systematized with references to support it. Pay attention to the technical details required to describe the experimental procedures.

5-     Authors just added the results without solid discussions. The discussion of results needs major revision. It should be very thorough.

6-     Discussion of results is weak. It is suggested to compare the results of the present study with some similar studies. More explanations and interpretations must be added for the results.

7-     In my opinion, authors should add new idea to this type of studies. The novelty and potential impact of this paper should be highlighted more.  

8-     It is suggested that all the tables be designed as three-line tables. Make all tables in the standard format of Table presentation. The Tables should not have all vertical lines hidden with most horizontal lines also hidden leaving only Top and bottom horizontal lines.

9-     Reduce the word count of the conclusions. It is advised to remove all paragraphs. The section must be improved with more pertinent conclusions.

10-   The references are not enough to justify the work done. Kindly improve on that by providing related, recent, and updated references.

Comments on the Quality of English Language

Moderate English correction required

Author Response

Dear reviewer #2,

I am pleased to submit the revised version of an original research article entitled “The impact of climate change on the water systems of the Yesil River basin in Northern Kazakhstan" for consideration for publication in Sustainability..

We have carefully updated the content of the manuscript following the comments of the four reviewers. This updated version of the manuscript takes into account all the suggestions made by the reviewers of the original manuscript. The new version of the modifies the requested sections of the original one. The manuscript has been reorganized and more references have been included. Some figures have been replotted as requested by some of the reviewers.

While the original version of the manuscript was 7975 words and included 28 references, this updated version is 9003 words long and includes 37 references. It is not possible to mark every change done in the English use inside manuscript, as it has been fully revised. Please activate the “track-changes” option in word to visualize all these changes.

We have provided answers to every reviewer’s comment which have been sent to each reviewer through MDPI online platform. Once all the comments of the reviewers have been addressed, we believe that this updated version of the manuscript fulfills all their requirements and is now appropriate for publication in Water.Please find below the answers to your specific comments. 

Comments reviewer #2

1-     The introduction is too long. The section should be restructured in a way that clearly conveys the overall objective of the research. The goals should be restructured into general and specific objectives.

The introduction section has been rewritten and paragraphs have been restructured. General and specific goals are now clearly identified.

2-     Most paragraphs in the entire manuscript, especially in the introduction section, are either too lengthy or too short. Kindly consider at least 4 and at most 9 sentences in each paragraph. It is suggested to revise all through the manuscript.

The whole manuscript has been rewritten as suggested.

3-      The novelty and practical applicability of this study should be highlighted more in the introduction section. The introduction section could be improved. Author should refer to these articles and several others relating to the subject matter from reputable journals to enrich and enhance the introduction and other sections of the manuscript. These are: https://doi.org/10.1007/s00704-023-04466-5

The scientific novelty of this work is now clearly marked at the last paragraph of the introduction section. The suggested reference has been included too, together with other new references.

4-     The paper needs more effort in displaying the methodology as the applied is not so innovative. It lacks information of different indices used in this research. This section should be re-organized, more specific and systematized with references to support it. Pay attention to the technical details required to describe the experimental procedures.

The methodology section has been restructured and new paragraphs have been added as requested. The whole structure of the manuscript has been revised. 

5-     Authors just added the results without solid discussions. The discussion of results needs major revision. It should be very thorough. ---

New paragraphs have been added to the discussion section as requested

6-     Discussion of results is weak. It is suggested to compare the results of the present study with some similar studies. More explanations and interpretations must be added for the results.

This paper is one of the scarce scientific contributions related with climate of the Yesil river basin. It is not posible to discuss the results by comparing them with analog studies.

7-     In my opinion, authors should add new idea to this type of studies. The novelty and potential impact of this paper should be highlighted more.

Тhe novelty of this research has been highlighted in the introduction section

8-     It is suggested that all the tables be designed as three-line tables. Make all tables in the standard format of Table presentation. The Tables should not have all vertical lines hidden with most horizontal lines also hidden leaving only Top and bottom horizontal lines.

The tables have been revised and adapted to the journal's format

9-     Reduce the word count of the conclusions. It is advised to remove all paragraphs. The section must be improved with more pertinent conclusions.

The conclusions section has been fully rewritten as requested, following the comment of all the reviewers.

10-   The references are not enough to justify the work done. Kindly improve on that by providing related, recent, and updated references.

New references have been provided following the advise of all the reviewers

Reviewer 3 Report

Comments and Suggestions for Authors

The manuscript entitled The impact of climate change on the water systems of the Yesil River basin in Northern Kazakhstan after Major  revision

The Abstract

This section could be expanded a bit. In this section, the authors only need to summarize the main results of this research in two sentences. In this section, the main findings of the manuscript need to be added.

The Keywords section needs to be changed in some way; at least one word needs to be added to this section that better explains the methods used in this research. For example, add a word that can explain the methodology.

Introduction

In lines 30 and 43, the authors give many important sentences about CO2 and coal distribution at the global level, but the number of references is very small. So the authors need to add more references.

The same is true for lines 40 and 61, where many sentences are given but no supporting documents and no references are given.

In this part of the manuscript, the authors need to describe more about the geographical location of Kazakhstan and the general drainage (water system) of the country. Also in this part the already published research papers on the same or similar topics are welcome.

Materials and Methods

Lines 174, 175 M.I. Lvovich is one of the greatest hydrologists of the SSSR, but the authors need to better describe why they analyzed his equations. I know that his equations describe the water balance in an area. So explain better.

Figure 1, this map must have mandatory geographic coordinates and a north arrow.

Table 1,

It is good to have the elevation data.

Materials and methods section is very short and it is mandatory to expand it.

It is not clear what method will be used to analyze the hydrologic characteristics of the river or river basin being studied.

In this section of the manuscript, I strongly recommend that the authors read and cite valuable references.

- Valjarević, A., Popovici, C., Štilić, A. et al. Cloudiness and water from cloud seeding in connection with plants distribution in the Republic of Moldova. Appl Water Sci 12, 262 (2022). https://doi.org/10.1007/s13201-022-01784-3.

- Karakus P. Investigation of Meteorological Effects on Çivril Lake, Turkey, with Sentinel-2 Data on Google Earth Engine Platform. Sustainability. 2023; 15(18):13398. https://doi.org/10.3390/su151813398.

Results

This section is too large and needs to be divided into two parts.

The first part could be labeled "Results of the meteorological stations" (or similar)

Second Numerical analysis (or statistical)

Line 371

This part of the manuscript is in contrast to the other parts. In this part it is good to add a specific title. This title could be Socioeconomic Factors.

It is also not clear to me how the authors measured the sector water supply % without statistical or GIS; remote sensing analysis, please explain better.

Discussion

In this section, the authors have to add more research results already published.

Also, the advantages and disadvantages of this research are explained.

Conclusion

In this section, the authors need to find answers to the following questions?

Why is this research important?

In one sentence, explain a better methodology for this research

How did the authors measure and estimate climate change impacts on river basins?

I recommend a comprehensive revision

The paper is good and scientifically accurate

The number of references is very low and more new references to this research should be included.

Good luck to the authors

Reviewer#2

Author Response

Dear reviewer #3,

I am pleased to submit the revised version of an original research article entitled “The impact of climate change on the water systems of the Yesil River basin in Northern Kazakhstan" for consideration for publication in Sustainability..

We have carefully updated the content of the manuscript following the comments of the four reviewers. This updated version of the manuscript takes into account all the suggestions made by the reviewers of the original manuscript. The new version of the modifies the requested sections of the original one. The manuscript has been reorganized and more references have been included. Some figures have been replotted as requested by some of the reviewers.

While the original version of the manuscript was 7975 words and included 28 references, this updated version is 9003 words long and includes 37 references. It is not possible to mark every change done in the English use inside manuscript, as it has been fully revised. Please activate the “track-changes” option in word to visualize all these changes.

We have provided answers to every reviewer’s comment which have been sent to each reviewer through MDPI online platform. Once all the comments of the reviewers have been addressed, we believe that this updated version of the manuscript fulfills all their requirements and is now appropriate for publication in Water.Please find below the answers to your specific comments. 

Comments reviewer #3

The Abstract

This section could be expanded a bit. In this section, the authors only need to summarize the main results of this research in two sentences. In this section, the main findings of the manuscript need to be added. The Keywords section needs to be changed in some way; at least one word needs to be added to this section that better explains the methods used in this research. For example, add a word that can explain the methodology.

The full abstract section has been updated, as also requested by other reviewers. The list of keywords have been updated

Introduction

In lines 30 and 43, the authors give many important sentences about CO2 and coal distribution at the global level, but the number of references is very small. So the authors need to add more references. The same is true for lines 40 and 61, where many sentences are given but no supporting documents and no references are given. In this part of the manuscript, the authors need to describe more about the geographical location of Kazakhstan and the general drainage (water system) of the country. Also in this part the already published research papers on the same or similar topics are welcome.

The whole introduction section has been rewritten. New references have been provided and the reviewers comments have been addressed. 

Materials and Methods

Lines 174, 175 M.I. Lvovich is one of the greatest hydrologists of the SSSR, but the authors need to better describe why they analyzed his equations. I know that his equations describe the water balance in an area. So explain better.

These data were used only to classify the type and mode of feeding of the Yesil River.

Figure 1, this map must have mandatory geographic coordinates and a north arrow.

Figure 1 has been updated

Table 1, It is good to have the elevation data.

Table 1 has been updated

Materials and methods section is very short and it is mandatory to expand it.

This section has been expanded has requested by the reviewers

It is not clear what method will be used to analyze the hydrologic characteristics of the river or river basin being studied.

We took into account your remark. New paragraphs have been added to explain the methodology

In this section of the manuscript, I strongly recommend that the authors read and cite valuable references.

The second reference you provide has been added as suggested, as it shows interesting relations with our research.

Results

This section is too large and needs to be divided into two parts. The first part could be labeled “Results of the meteorological stations” (or similar). Second Numerical analysis (or statistical). Line 371 This part of the manuscript is in contrast to the other parts. In this part it is good to add a specific title. This title could be Socioeconomic Factors.

All these suggestions have been taken into account.

It is also not clear to me how the authors measured the sector water supply % without statistical or GIS; remote sensing analysis, please explain better.

Data on the amount of water use in various sectors of the economy were provided by the Kazakh National Committee on Water Resources as explained in section 3.3 Socioeconomic factors

Discussion

In this section, the authors have to add more research results already published. Also, the advantages and disadvantages of this research are explained .

We took into account your remark. New references have been provided.

Conclussion

In this section, the authors need to find answers to the following questions?

Why is this research important?  In one sentence, explain a better methodology for this research  How did the authors measure and estimate climate change impacts on river basins?

The conclusion section has been rewritten, taking into account your considerations.

The number of references is very low and more new references to this research should be included.

New references have been provided. Now the paper is completed with 37 references.

Reviewer 4 Report

Comments and Suggestions for Authors

First, I was expecting after collecting all of this information related to temperature, water consumption, population, etc. to do a correlation of to do prediction of what is expected to happen in the future. You can find other comments below:

*Abstract: the main findings/results of your research are not presented.

increasing temp. by approximately 0.7 °C and precipitation by 5.5%”, these not a results of your research. What are the main results you have obtained based on the historical data?

*Provide reference for the following: lines 32-34, lines 38-40, lines 41-43, lines 44-48, lines 51-56, lines 57-63, lines 91-93, lines 98-99, line 102-105, lines 115-117, lines 176-178, lines

*Lines 107-108: “In an extreme scenario characterized by high GHG emissions, there is a concerning projection that the wet zone will shift northward by 250-300 km by 2085.” Who done this projection? Provide suitable reference.

*Introduction section needs improvement: (1) there are many information without sufficient reference, (2) where is the previous research in which you have built the methodology of your study. You have to mention examples of already published papers that are close to your topic.

*Line 197: remove the typo.

*Table 1: what the symbol above IBMP Burabay refer to?    “ * ”

*Is this your results?: “These increases are projected to range from 1.2 °C 426 in the near future to as high as 6.4 °C in the distant future under various emission scenarios. Of note, the Zhabai River catchment area is expected to experience the most substantial temperature rise, with projections of 3.9 °C under one scenario and 6.4 °C under an-429 other by the end of the century [20].

Comments on the Quality of English Language

 Moderate editing of English language required.

Author Response

Dear reviewer #4,

I am pleased to submit the revised version of an original research article entitled “The impact of climate change on the water systems of the Yesil River basin in Northern Kazakhstan" for consideration for publication in Sustainability..

We have carefully updated the content of the manuscript following the comments of the four reviewers. This updated version of the manuscript takes into account all the suggestions made by the reviewers of the original manuscript. The new version of the modifies the requested sections of the original one. The manuscript has been reorganized and more references have been included. Some figures have been replotted as requested by some of the reviewers.

While the original version of the manuscript was 7975 words and included 28 references, this updated version is 9003 words long and includes 37 references. It is not possible to mark every change done in the English use inside manuscript, as it has been fully revised. Please activate the “track-changes” option in word to visualize all these changes.

We have provided answers to every reviewer’s comment which have been sent to each reviewer through MDPI online platform. Once all the comments of the reviewers have been addressed, we believe that this updated version of the manuscript fulfills all their requirements and is now appropriate for publication in Water.Please find below the answers to your specific comments. 

Comments reviewer #4

*Abstract: the main findings/results of your research are not presented.

The abstract section has been rewritten, as requested by the reviewers

*“increasing temp. by approximately 0.7 °C and precipitation by 5.5%”, these not a results of your research. What are the main results you have obtained based on the historical data?

These paragraphs have been rewritten for a better understanding

*Provide reference for the following: lines 32-34, lines 38-40, lines 41-43, lines 44-48, lines 51-56, lines 57-63, lines 91-93, lines 98-99, line 102-105, lines 115-117, lines 176-178, lines

References have been provided in the manuscript

*Lines 107-108: “In an extreme scenario characterized by high GHG emissions, there is a concerning projection that the wet zone will shift northward by 250-300 km by 2085.” Who done this projection? Provide suitable reference.

References have been provided in the manuscript

*Introduction section needs improvement: (1) there are many information without sufficient reference, (2) where is the previous research in which you have built the methodology of your study. You have to mention examples of already published papers that are close to your topic.

References have been provided in the manuscript

*Line 197: remove the typo.

Typo has been removed

*Table 1: what the symbol above IBMP Burabay refer to?

IBMP refers to “Integrated Background Monitoring Post” as it is now shown at the foot of Table 1

*Is this your results?: “These increases are projected to range from 1.2 °C 426 in the near future to as high as 6.4 °C in the distant future under various emission scenarios. Of note, the Zhabai River catchment area is expected to experience the most substantial temperature rise, with projections of 3.9 °C under one scenario and 6.4 °C under an-429 other by the end of the century [20].

This is not the result of our research work, we used the results of the following authors Iulii Didovets, etc., according to the link given.

Round 2

Reviewer 1 Report

Comments and Suggestions for Authors

Accept in present form

Author Response

Thanks very much for your review. 

Reviewer 2 Report

Comments and Suggestions for Authors

Authors have significantly improved the revised manuscript 

Author Response

Thanks very much for your review

Reviewer 3 Report

Comments and Suggestions for Authors

The manuscript entitled The impact of climate change on the water systems of the Yesil River basin in Northern Kazakhstan after Minor (Moderate) Revision.

I have more concerns in this manuscript, even if the authors improve this text.

The main concerns are still related to these parts

It is also not clear to me how the authors measured the % water supply to the sector without statistical or GIS or remote sensing analysis. Climate change impacts are not adequately presented in this manuscript, including for the Kazakhstan area. Climate impacts are also very important for the river basin.

The references to climate change and climate in general are still insufficient. Therefore, I strongly recommend the authors to read and cite a valuable reference.

- Valjarević, A., Milanović, M., Gultepe, I., Filipović, D. & Lukić, T. (2022) Updated Trewartha climate classification with four climate change scenarios. The Geographical Journal, 188, 506–517. Available at: https://doi.org/10.1111/geoj.12458.

Discussion

In this section, the authors have to add more research results already published. Also, the advantages and disadvantages of this research are explained.

Conclusion

In this section, the authors need to find answers to the following questions?

Why is this research important? Explain in one sentence a better methodology for this research How did the authors measure and estimate climate change impacts on river basins?

The number of references is very low and more new references to this research should be included.

The paper has the potential to be published, but after a minor (moderate) revision

 Good luck to the authors

Reviewer#3

Author Response

Dear reviewer #3,

I am pleased to submit the revised version of an original research article entitled “The impact of climate change on the water systems of the Yesil River basin in Northern Kazakhstan" for consideration for publication in Sustainability.

We have carefully updated the content of the manuscript following the comments of the four reviewers. This updated version of the manuscript takes into account all the suggestions made by the reviewers of the original manuscript. The new version of the modifies the requested sections of the original one. The manuscript has been reorganized and more references have been included. Some figures have been replotted as requested by some of the reviewers. Please activate the “track-changes” option in word to visualize all these changes.

We have provided answers to every reviewer’s comment which have been sent to each reviewer through MDPI online platform. Once all the comments of the reviewers have been addressed, we believe that this updated version of the manuscript fulfills all their requirements and is now appropriate for publication in Water. Please find below the answers to your specific comments. 

Comments reviewer #3

  1. It is also not clear to me how the authors measured the % water supply to the sector without statistical or GIS or remote sensing analysis. Climate change impacts are not adequately presented in this manuscript, including for the Kazakhstan area. Climate impacts are also very important for the river basin.

The manuscript includes a full paragraph after Table 6 with this information. Statistical data about water consumption by economic sectors is now included as Figure 7 and 8 for Akmola and North Kazakhstan regions, according to statistical data from 2000 to 2021.

  1. The references to climate change and climate in general are still insufficient. Therefore, I strongly recommend the authors to read and cite a valuable reference.

- Valjarević, A., Milanović, M., Gultepe, I., Filipović, D. & Lukić, T. (2022) Updated Trewartha climate classification with four climate change scenarios. The Geographical Journal, 188, 506–517. Available at: https://doi.org/10.1111/geoj.12458.

The suggested reference has been added, as well as some other ones in the introduction section. The manuscript now has 41 references.

  1. Discussion: In this section, the authors have to add more research results already published. Also, the advantages and disadvantages of this research are explained.

Based on the statistical data of recent years, an analysis of several important indicators related to the Ishim river basin was carried out. In particular, indicators of temperature, precipitation, water flow and water consumption, population in the region and consumption of Water Resources in various sectors of the economy were considered. Based on this, a comprehensive assessment was carried out. Climate scenarios and hydrological forecasts for future climate change were also referred to, and the state of future water supply in the region was analyzed. However, there are a number of shortcomings in the study. Due to the small number of analog studies on this topic, there was little opportunity to analyze the results of the study in comparison.

  1. Conclusion: In this section, the authors need to find answers to the following questions?

Why is this research important? Explain in one sentence a better methodology for this research

Since Kazakhstan is located in the central part of the temperate zone of the Eurasian continent, the climate is sharply continental, therefore, the country has a poorly developed river system, it is considered necessary to carry out such comprehensive studies to prevent future water resource scarcity.

How did the authors measure and estimate climate change impacts on river basins?

Based on the data of stations located in the Ishim river basin over the past 60 years, temperature and precipitation indicators were analyzed, the nature of climate change was determined, as well as several annual indicators of water loss of the Ishim River were analyzed, which were determined by the correlation relationship of the basin with temperature and precipitation.

  1. The number of references is very low and more new references to this research should be included.

The suggested reference has been added, as well as some other ones in the introduction section. The manuscript now has 41 references.

Reviewer 4 Report

Comments and Suggestions for Authors

Thanks for addressing most of my comments. One last comment, I am still not satisfied with the introduction section since the previous research in which you have built the methodology of your study are not clearly presented. 

Comments on the Quality of English Language

Minor editing of English language required.

Author Response

Dear reviewer #4,

I am pleased to submit the revised version of an original research article entitled “The impact of climate change on the water systems of the Yesil River basin in Northern Kazakhstan" for consideration for publication in Sustainability.

We have carefully updated the content of the manuscript following the comments of the four reviewers. This updated version of the manuscript takes into account all the suggestions made by the reviewers of the original manuscript. The new version of the modifies the requested sections of the original one. The manuscript has been reorganized and more references have been included. Some figures have been replotted as requested by some of the reviewers. Please activate the “track-changes” option in word to visualize all these changes.

We have provided answers to every reviewer’s comment which have been sent to each reviewer through MDPI online platform. Once all the comments of the reviewers have been addressed, we believe that this updated version of the manuscript fulfills all their requirements and is now appropriate for publication in Water. Please find below the answers to your specific comments. 

Comments reviewer #4

  1. 1. I am still not satisfied with the introduction section since the previous research in which you have built the methodology of your study are not clearly presented.

Four new references have been added. in the introduction section. The manuscript now has 41 references. New figures 7, 8 and 9 have been included too, together with new comments in the discussion section